# Impact of Hypoxia on Carbon Ion Therapy in Glioblastoma Cells: Modulation by LET and Hypoxia-Dependent Genes

**DOI:** 10.3390/cancers12082019

**Published:** 2020-07-23

**Authors:** Samuel Valable, Aurélie N. Gérault, Gaëlle Lambert, Marine M. Leblond, Clément Anfray, Jérôme Toutain, Karim Bordji, Edwige Petit, Myriam Bernaudin, Elodie A. Pérès

**Affiliations:** UNICAEN, CNRS, CEA, ISTCT/CERVOxy group, GIP Cyceron, Normandie Univ, 14000 Caen, France; valable@cyceron.fr (S.V.); gerault.a@gmail.com (A.N.G.); gaelle.lambert@laposte.net (G.L.); leblond.marine.m@gmail.com (M.M.L.); clement.anfray@humanitasresearch.it (C.A.); toutain@cyceron.fr (J.T.); bordji@cyceron.fr (K.B.); epetit@cyceron.fr (E.P.); bernaudin@cyceron.fr (M.B.)

**Keywords:** hypoxia, glioblastoma, heavy ion radiotherapy, carbon ions, erythropoietin

## Abstract

Tumor hypoxia is known to limit the efficacy of ionizing radiations, a concept called oxygen enhancement ratio (OER). OER depends on physical factors such as pO_2_ and linear energy transfer (LET). Biological pathways, such as the hypoxia-inducible transcription factors (HIF), might also modulate the influence of LET on OER. Glioblastoma (GB) is resistant to low-LET radiation (X-rays), due in part to the hypoxic environment in this brain tumor. Here, we aim to evaluate in vitro whether high-LET particles, especially carbon ion radiotherapy (CIRT), can overcome the contribution of hypoxia to radioresistance, and whether HIF-dependent genes, such as erythropoietin (EPO), influence GB sensitivity to CIRT. Hypoxia-induced radioresistance was studied in two human GB cells (U251, GL15) exposed to X-rays or to carbon ion beams with various LET (28, 50, 100 keV/µm), and in genetically-modified GB cells with downregulated EPO signaling. Cell survival, radiobiological parameters, cell cycle, and ERK activation were assessed under those conditions. The results demonstrate that, although CIRT is more efficient than X-rays in GB cells, hypoxia can limit CIRT efficacy in a cell-type manner that may involve differences in ERK activation. Using high-LET carbon beams, or targeting hypoxia-dependent genes such as EPO might reduce the effects of hypoxia.

## 1. Introduction

Glioblastoma (GB), also defined as a grade IV glioma based on the WHO classification, is the most common malignant primary brain tumor in adults, representing up to 46% of all gliomas and 16% of all primary brain tumors [1]. GB is one of the most aggressive tumors and has a poor prognosis, with a median overall survival from diagnosis of about 15–18 months and less than 10% of patients surviving more than five years [2]. The current standard of care for GB patients includes maximal tumor resection, postoperative external-beam radiation therapy (RT) (60 Gy in 2-Gy fractions) with concomitant temozolomide (TMZ, 75 mg/m^2^), followed by adjuvant TMZ (150–200 mg/m^2^) for six cycles. Unfortunately, tumor recurrence occurs early in the vast majority of patients (90%) [3]. Thus, treatment efficacy for GB clearly needs to be improved. However, the location of these tumors within the sensitive brain parenchyma along with their infiltrative and hypoxic patterns limit the use of aggressive local treatments, including conventional RT. It is known that GB is resistant to X-rays, in particular due to the presence of hypoxia in these brain tumors [4].

Over the last decades, several studies have proposed particle therapy as a new promising therapeutic approach in the management of GB patients [5,6]. As such, carbon ion radiotherapy (CIRT) is currently being clinically investigated for treatment of several malignant tumors, including GB [7,8,9]. Heavy-particle therapy, especially CIRT, offers two main advantages compared to conventional photon-based RT, i.e., a better ballistic accuracy, and a greater biological effectiveness. Carbon ions have distinct physical characteristics based on their inverted dose profile, and low doses of radiation are deposited within the entry channel of the beam, followed by a high dose deposition in the depth. This phenomenon, called the Bragg peak, can be used to precisely direct radiations into the defined lesion by varying the energy of the particle beam as the peak is followed by an extremely steep dose fall-off. Therefore, CIRT can precisely target the tumor tissues while sparing the surrounding healthy tissues, allowing to scale up the dose and reducing severe side effects [10]. Moreover, particles with high linear energy transfer (LET), which defines the energy loss along the ion path, exert a higher relative biological effectiveness (RBE) in tumor cells. RBE is a radiobiological parameter that compares the efficiency of different types of radiation relative to X-rays (low-LET radiation) in producing a defined biological effect. Low-LET radiation refers to gamma rays and X-rays, with LET values below 5 keV/µm for medical applications, whereas high-LET radiation includes alpha particles, protons, neutrons, and heavy charged particles with LET up to 600 keV/µm. It has been reported that RBE is not constant but depends on particle type and energy level. In general, RBE increases as a function of LET, and is estimated to be 2.5–3 for high-LET carbon ions [11]. The greater RBE associated with carbon ions is due to more complex DNA damage, especially clustered DNA double-strand breaks, which are refractory to repair and can stimulate apoptosis.

Hypoxia, defined as the reduction of tissue oxygen levels, is a major feature of solid tumors, including GB [4]. Hypoxia correlates with poor patient outcome, aggressive tumor phenotype, and resistance to chemotherapy and radiotherapy in GB [12]. Indeed, hypoxia-induced radioresistance is mainly attributed to the “oxygen effect” [13,14]. In the presence of molecular oxygen and following RT, DNA can be damaged directly by ionizing radiation or indirectly through formation of reactive oxygen species (ROS) via water radiolysis, enzymatic production or altered aerobic metabolism. Under hypoxic conditions, there is less DNA damage due to a decrease in radiation-induced oxidative stress and a better efficiency of DNA repair systems [15]. Thus, cell killing is greater under normoxic conditions compared to hypoxic conditions, giving rise to the concept of oxygen enhancement ratio (OER). OER greater than one indicates the presence of hypoxia-dependent radioresistance. Previous in vitro studies with low-LET radiation have shown that tumor cells are two to three times more radioresistant under hypoxic than normoxic conditions, whereas OER can reach one with high-LET radiation [16]. Interestingly, OER is inversely correlated with LET, suggesting a potential clinical advantage of high-LET radiotherapy with heavy ion beams such as CIRT compared to low-LET photon or proton irradiation [17,18].

In addition to these physicochemical mechanisms of irradiation, various biological parameters may also affect CIRT efficiency. This is the case of the cellular adaptive responses to hypoxia mediated by the activation of the hypoxia-inducible transcriptional factor-1 (HIF-1) [19]. HIF-1 has been suggested to be involved in radioresistance under hypoxic conditions by acting on multiple cellular pathways. Indeed, HIF-1 and its derived genes are involved in the survival of tumor cells by modulating glucose metabolism, which leads to overproduction of antioxidants, regulation of the cell cycle, activation of DNA repair pathways, inhibition of apoptosis, and maintenance of cancer cell stemness [20,21]. Studies have also shown that HIF-1 inhibition sensitizes GB cells to X-rays [22,23]. Furthermore, the involvement of hypoxia in radioresistance has also been indirectly demonstrated by the modulation of HIF-dependent gene expression such as EPO (erythropoietin) [24], VEGF (vascular endothelial growth factor) [25], GLUT-1 (glucose transporter 1) [26], or DNA-PK (DNA-dependent protein kinase) [27]. Although the modulation of HIF-1 expression after exposure to carbon ions has been evaluated [28,29], the influence of HIF-1 and/or HIF-dependent genes on the intrinsic sensitivity of tumor cells to CIRT is poorly studied [30,31].

The aim of this in vitro study was to determine the impact of oxygen levels on the sensitivity of GB cells to CIRT. We used human GB cell lines to determine: (i) whether carbon ion irradiation can overcome hypoxia-induced radioresistance; and (ii) whether varying LET values or downregulating the EPO receptor (EPOR) can modulate GB cell sensitivity to carbon ions under hypoxic conditions.

## 2. Results

### 2.1. The Relationship between Radiobiological Effects of Carbon Ions and LET on GB Cells

Although many studies have shown that the RBE of carbon ions enhances with LET in many types of solid tumors, only a few addressed this issue in GB. Thus, we first investigated the biological response of GB cells to CIRT at LET values ranging from 28 to 100 keV/µm to approximate those used in the clinics (Appendix A). Indeed, it has been shown that the optimum LET to maximize the therapeutic advantage on tumors is between 25 to 75 keV/µm, while minimization of the effects to normal tissues [32]. The maximal LET must not exceed 100 keV/µm because over this value, the RBE of carbon ions drops rapidly as the dose in excess not necessary for cell killing is transferred to a cell in the track of a single ion (phenomena called the “overkill effect”) [33]. In this study, 28 keV/µm (defined as low-LET CIRT) corresponds to the energy deposit upstream of the Bragg peak, 50 keV/µm refers to a deposit energy at the beginning of the Bragg peak and 100 keV/µm (corresponds to high-LET CIRT) is the deposit energy maximal (without induce overkill effect).

After exposing U251 cells to CIRT, we observed that the colony counts decreased with increasing doses of CIRT, especially at high-LET values (Figure 1A). As expected, GB cells were more sensitive to CIRT than to conventional X-ray irradiation at all LET, as evidenced by survival curves (Figure 1B) and radiobiological parameter measurements (Figure 1C). In comparison to photons, exposing U251 cells to CIRT at the maximum LET value of the Bragg peak (100 keV/µm) led to four times lower cell survival at 2 Gy (*p* < 0.0001) (Figure 1C). Interestingly, the GB cell sensitivity to CIRT significantly increased with increasing LET values (Figure 1C). Thus, RBE was strongly, linearly, and positively correlated to LET (r^2^ = 0.99) (Figure 1D), confirming that U251 GB cell sensitivity to CIRT is a function of LET. 

In order to better understand the response of GB cells to CIRT as a function of LET, we studied the cell cycle of U251 cells at an early time point post-CIRT (14 h) to detect cell cycle arrest and at a later time (72 h) to assess irradiation-induced cell death (Figure 2). From the cell cycle profiles, we observed at 14 h that CIRT induced a G2/M arrest at all LET values in U251 cells (Figure 2A,B), which preceded an increase in cell number in the subG1 phase at 72 h, reflecting radiation-induced apoptosis (Figure 2A,C). However, the G2/M arrest was less pronounced with high-LET as the proportion of U251 cells in G2/M at 14 h post-CIRT was 66% and 55% with LET of 28 and 100 keV/µm, respectively (*p* < 0.01) (Figure 2B). This effect is likely due to a smaller proportion of U251 cells remaining in the G0/G1 phase at the highest LET value. A similar increase in the proportion of GB cells in the subG1 phase was also observed 72 h after CIRT at any LET values (around 30% for the irradiated cells compared to 9% for the control cells). It is to be noted that a G2/M arrest was always present 72 h post-CIRT at 100 keV/μm. This effect may indicate more deleterious cell damage in GB cells exposed to carbon ions with high-LET (Figure 2C). Therefore, these data show that the biological effectiveness of CIRT on GB cells results in an LET-dependent G2/M arrest, followed by GB cell accumulation in the subG1 phase.

### 2.2. Effects of Hypoxia on GB Sensitivity to Carbon Ion Irradiation as a Function of Cell Lines and LET

In radiobiology studies of heavy ion particles, it is postulated that the oxygen effect does not affect the tumor cell response to irradiation. However, only a few studies have tested this concept, in particular in GB, a brain tumor known to be highly hypoxic. Thus, we investigated the efficacy of CIRT under hypoxic conditions (1% O_2_) in U251 cells and in GL15, another human GB cell line. As shown in Figure 3A,B, both cell lines were more resistant to X-ray exposure under hypoxic than normoxic conditions (21% O_2_) (*p* < 0.05). These results were confirmed by quantification of the radiobiological parameters. The OER value was greater than 1, indicating the presence of hypoxia-dependent resistance to X-rays, for both GB cell lines (Figure 3C,D). In contrast, the radiosensitivity of these cells to CIRT appeared to be a function of the oxygen levels. As seen in the images acquired at 72 h after carbon ion exposure (4 Gy), U251 cell density appeared similar after irradiation under normoxic and hypoxic conditions (Figure 3A), but GL15 cell density was higher after irradiation under hypoxia than normoxia (Figure 3B). After CRIT exposure of U251 cells, the RBE values were 1.5 ± 0.1 under normoxia and 1.8 ± 0.1 under hypoxia (Figure 3C) and no oxygen effect was observed (OER = 1.0 ± 0.0). For the GL15 cells, although carbon ions tended to be more effective than X-rays under normoxia (RBE = 1.3 ± 0.2), the effect was not as obvious under hypoxia (RBE = 1.2 ± 0.4) (Figure 3D). The radiobiological parameters (SF2, D10, and D37) were significantly different between normoxic and hypoxic conditions in GL15 cells, suggesting an oxygen effect of carbon ion irradiation. More importantly, unlike U251 cells, GL15 cells exhibited radioresistance to carbon ions under hypoxia with a significant OER (*p* < 0.05 versus theoretical value of 1) (Figure 3D).

Next, we investigated the potential underlying mechanisms of the differential response to CIRT of these two GB cell lines under hypoxia. The ERK pathway is known to be involved in hypoxia-induced radioresistance to X-rays [34,35], and to be modulated by carbon ions [36,37]. Thus, we tested the activation of ERK (P-ERK) using western blotting and phospho-specific antibodies. Hypoxia alone led to ERK1/2 phosphorylation in both cell lines (Figure 4A,B). CIRT did not modulate ERK phosphorylation under either oxygenation conditions in U251 cells (Figure 4A). In contrast, CIRT under hypoxia induced a significant increase in ERK activation after 24 h in GL15 cells compared to control cells (*p* < 0.05) or CIRT-exposed cells under normoxia (*p* < 0.01) (Figure 4B). These results suggest that the hypoxia-induced, cell type-dependent radioresistance to CIRT might be partly explained by differences in ERK activation.

The oxygen effect of heavy ions is reduced by high-LET in various tumors [16,18]. As the survival curves for U251 cells did not depend on oxygenation or LET, suggesting that U251 cell sensitivity to CIRT is independent of an oxygen effect (Appendix A), we wanted to test these effects in GL15 cells that display hypoxia-induced radioresistance to CIRT. As presented in Figure 5A, the OER values in GL15 cells decreased with increasing LET (OER = 1.4, 1.3, and 1.0 for CIRT with LET of 28, 50, and 100 keV/µm, respectively). Based on the negative linear correlation (r^2^ = 0.99) between OER and LET (Figure 5B), particles with very high-LET might be of interest to overcome the oxygen effect in response to CIRT. 

### 2.3. Impact of Erythropoietin on the CIRT Efficacy in GB Cells

EPO, an HIF-1-regulated gene, has previously been shown to be involved in X-ray radioresistance [24]. Thus, we studied the effect of knocking down the EPO receptor (EPOR) on GB cell sensitivity to CIRT (50 keV/µm) under normoxia. In these experiments, we chose to use an LET value of 50 keV/µm in order to match the optimum LET range used in clinical practice (25–75 keV/µm) and to avoid the overkill effect usually associated with LET values > 100 keV/µm. As shown in Figure 6A, inhibition of EPOR expression on U251 cells led to decreased cell survival compared to scrambled shRNA cells (*p* < 0.01). The quantification of the radiobiological parameters confirmed that EPOR silencing induces the radiosensitization of U251 cells to CIRT as evidenced by the SER (sensitization enhancement ratio, defined as ratio D37 scrambled cells/D37 shEPOR cells) equivalent to 1.3 ± 0.1 (*p* < 0.05, compared to the theoretical value 1). In addition to the cell survival decrease, at high doses (4 and 6 Gy) of CIRT, the colony size of U251-shEPOR cells was smaller relative to that of the control cells (Figure 6B). In parallel, we studied the effects of CIRT combined with EPOR downregulation on the cell cycle of U251 cells (Figure 6C). First, we noticed that EPOR knockdown alone induced a significant G2/M arrest at the expense of the G0/G1 phase. At 14 h post-CIRT, the U251-shRNA scrambled cells were in G2/M arrest, and the G2/M arrest was potentiated by EPOR silencing (*p* < 0.05) (Figure 6C). Similarly, at 72 h after CIRT, a greater proportion of U251 shRNA-scrambled cells was in subG1 phase, an effect that was amplified by inhibition of EPOR expression (Figure 6C). These results indicate that EPOR knockdown radiosensitizes U251 cells to CIRT by increasing apoptosis (subG1 phase) consecutive to an increase in DNA damage (G2/M arrest). 

## 3. Discussion

Although CIRT would be a relevant strategy for treating hypoxic tumors, studies on its effects for GB treatment remain sparse, justifying further experimental studies to evaluate the biological effectiveness of this heavy ion radiation therapy as well as the role of the oxygen effect. In this in vitro study, we confirm previous studies showing that CIRT is more effective than conventional RT with X-rays in human GB cells [38,39]. Furthermore, we show in two human GB cell lines (U251 and GL15) that RBE is strongly correlated with LET (ranging from 28 keV/µm to 100 keV/µm), but is cell-type dependent. These results are in agreement with those from Chew and colleagues who reported similar findings in other human GB cell lines (U87, T98G and LN18) [40]. Our data also highlight that the RBE increase is associated with a delay in G2/M arrest at low-LET (28 keV/µm), and with a prominent G2/M blockage at high-LET (100 keV/µm). This early cell cycle arrest is followed by apoptosis, the intensity of which depends on LET [41], but is independent on the p53 status^44^. A cell cycle arrest at a checkpoint indicates DNA damage detection and DNA repair activation. However, in GB cells, the potential lethal damage repair (PLDR) is decreased with high-LET values [42]. Thus, our results indicate that the RBE of carbon ions is mainly a reflection of high-LET, which produces more complex DNA damage than photons and low-LET carbon ions, especially inducing clustered DNA double-strand breaks that are refractory to repair, leading to GB cell death by apoptosis. Increased apoptosis with high-LET values has been demonstrated by the nuclear fragmentation observed in the subG1 phase of the cell cycle, but analyses of protein expression of cleaved caspase-3 and cleaved PARP (Poly (ADP-Ribose) Polymerase) or annexin-V IP studies by cytometry under these conditions would be of interest to further support these conclusions. 

CIRT is generally believed to overcome hypoxia-dependent radioresistance. However, our results in GB cells show that this assumption is not binary as we determined an OER = 1 for U251 GB cells exposed to carbon ions with a LET of 28 keV/μm, but an OER around 1.4 for GL15 cells. The oxygen effect is therefore cell type-dependent and could be due to a difference in ERK activation between these GB cells. To clearly demonstrate the role of the ERK pathway in hypoxia-induced radioresistance, complementary experiments with inhibitors of this pathway, such as U0126, would be necessary. However, our hypothesis of the involvement of ERK signaling in hypoxia-induced carbon ion irradiation is supported by the recent work by Tomiyama et al., highlighting that knockdown of MEK 1/2 or the use of MEK-specific inhibitors improve CIRT response of GB cells [43]. Indeed, this study demonstrated that the efficacy of carbon ions is associated with a sustained inhibition of ERK activation and a crucial role of the MEK-ERK cascade in the death of GB cells exposed to carbon beam [43]. In contrast, the activation of the MEK-ERK pathway plays a role in X-ray GB radioresistance [34,36]. In our study, we focused on ERK activation to investigate the mechanisms underlying hypoxia-induced resistance to CIRT, but we cannot rule out the possible involvement of other signaling pathways. Thus, future experiments using large-scale genomic and/or proteomic studies could help differentiate the U251 and GL15 responses to carbon ion irradiation, especially in hypoxia.

As described in several cell lines from various tumor types, we also observed that the OER is also inversely correlated to the LET in GB cells (ranging from 28 to 100 keV/µm). This is the case for GL15 GB cells, for which the OER value decreases to 1 when these cells were exposed to 100 keV/µm carbon ions. However, although attractive, such a strategy based on LET increase to overcome hypoxia-dependent radioresistance could be limited by the overkill phenomenon.

Alternatively, it might be relevant to target a hypoxia-dependent pathway to reduce GB resistance to carbon ions. Among these pathways, the transcription factor HIF-1, the main regulator of cellular oxygen homeostasis and cell survival, is a potential target. Indeed, HIF-1 is overexpressed in the hypoxic tissues of tumors, but also in response to different oncogenes, different signaling pathways and stress condition such as radiations. HIF-1 is strongly implicated in the resistance of GB cells to X-rays [22,23]. However, although several studies sought to delineate the HIF signaling in response to carbon ion radiation, few have focused on establishing the importance of HIF in the response of cancer cells to CIRT, in particular in GB cells [28,29,44]. In these aggressive brain tumors, HIF-1 inhibition in combination with CIRT would be an attractive strategy as it might alleviate hypoxia-induced radioresistance and tumor recurrence due to the radioresistance of the GB stem cells. Nevertheless, specifically targeting HIF-1 remains a challenge.

Another strategy to diminish hypoxia-dependent radioresistance would be to target HIF-regulated genes. Accordingly, it was recently proposed in human lung cancer models to combine CIRT with an inhibitor of the DNA-PK. This HIF-dependent gene is one of the key players in the DNA damage response to complex double-strand breaks [45]. In these tumor cells, a radiosensitizing effect of a DNA-PK inhibitor was observed in normoxia and hypoxia. In our study, we chose to evaluate the impact of EPO, another well-known HIF-1-regulated gene. In a previous study, we demonstrated that inhibition of EPO/EPOR signaling in GB cells strengthens X-ray efficacy [24]. In the present study, we show that EPOR silencing radiosensitizes GB cells to CIRT by intensifying the apoptosis consecutive to increased DNA damage. Our results, in accordance with our previous study, also suggest that the improvement of GB responsiveness to CIRT could be due to an attenuation of ERK activation, as we have already shown that EPOR silencing reduces ERK phosphorylation [46]. These results were obtained under normoxia, and it would be interesting in future experiments to investigate the involvement of EPO/EPOR signaling in hypoxia-induced resistance to carbon ion irradiation. It would also be suitable to evaluate the effects of EPOR inhibition in GL15 cells, a cell line that exhibits hypoxia-dependent radioresistance to CIRT.

## 4. Materials and Methods

### 4.1. Cell Lines

Two human glioblastoma-derived cell lines were used. U251-MG cells (Cellosaurus CVCL_0021) (wild-type IDH1/2, mutated p53, EGFRvIII, mutated PTEN) [47] were obtained from the National Cancer Institute (NCI, Besthesda, MD, USA). GL15 cells (Cellosaurus CVCL_5H95) (wild-type IDH1/2, mutated p53, EGFR amplification by the presence of 7–8 exons extra copied of chromosome 7, loss of chromosome 10 suggesting lack of PTEN expression) [48] were kindly given by JS Guillamo (CHU Caen, France). These cells were cultured in DMEM (Dulbecco’s Modified Eagle’s Medium) supplemented with 10% fetal calf serum, 2 mM glutamine and 100 U/mL penicillin/streptomycin. Both cell lines were maintained in culture at 37 °C with 5% CO_2_ and 95% humidity and seeded weekly.

Stable inhibition of EPOR expression in U251-MG cells was performed by shRNA interference brought by lentiviral particles (Sigma-Aldrich, Saint-Quentin Fallavier, France) and characterized in a previous study [46]. Control cells consisted of scrambled-shRNA infected cells.

### 4.2. Hypoxia Treatment

Hypoxia experiments were performed in a hypoxia workstation (IN VIVO 2 500, Baker Ruskinn, Alliance Bio Expertise) set at 1% O_2_, 5% CO_2_, and 94% N_2_ at 37 °C in humidified atmosphere. Culture medium was equilibrated for 30 min with the gas mixture before being added to the cells and incubated in the hypoxia chamber. The GB cells were maintained in these hypoxic conditions for 6 h before radiation exposure (i.e., 18 h after cell seeding) and maintained under hypoxic (1% O_2_) or normoxic conditions (21% O_2_) until the end of the experiment.

### 4.3. Radiation Procedure

X-ray treatment: After seeding and 24 h culture, cells were exposed at room temperature to X-rays at doses ranging from 0 to 8 Gy (X-Rad 225Cx, Precision *X*-ray Inc, CYCERON platform). X-rays were delivered at a mean energy of 80 keV at a dose rate of 2 Gy/min (voltage: 225 kV, current: 13 mA, Cu filter: 1 mm). Just after irradiation, culture medium was replaced with fresh equilibrated medium and cells were incubated under normoxic or hypoxic conditions until the end of the experiment.

Carbon ion treatment: Cells were exposed at room temperature to carbon ion beam (^12^C) with physical doses ranging 0 to 6 Gy (IRABAT D1, GANIL, Caen, France) at the energy of 95 MeV/u [49]. As presented in Appendix A, to generate carbon ions with various LET, a degrader (increasing thicknesses of PMMA (polymethyl methacrylate)) was placed upstream of the culture flasks, e.g., 28 keV/µm (PMMA = 0), 50 keV/µm (PMMA = 13.9 mm), and 100 keV/µm (PMMA = 17.9 mm) (Appendix A). To confirm the delivered doses for each culture flask, dose deposit maps of carbon ions were generated by an ionisation chamber-based dosimetry (DOSION III, LPC laboratory, Caen, France) (Appendix A). Considering the GANIL (Grand Accélérateur National d’Ions Lourds, Caen, France) facilities, cells were vertically irradiated requiring full filling of the flasks with culture media during the irradiation time. Just after carbon ion irradiation, the culture medium was removed and replaced by fresh equilibrated medium. The cells were maintained in normoxic or hypoxic conditions until the end of the experiment.

### 4.4. Cell Survival Analysis

Clonogenic survival assay: Cell survival was assessed by the clonogenic capacity of a single cell. All experiments were conducted in exponentially growing cells that were seeded 24 h before irradiation at increasing cell densities, as a function of the irradiation dose (200 to 1200 cells/mL). After exposure to X-rays or carbon ions, cells were incubated under normoxic or hypoxic conditions during 10 days. Then, colonies were stained with 1% crystal violet (Sigma-Aldrich) diluted in 20% ethanol. All colonies containing 50 cells or more were manually counted and considered as cells with unaffected clonogenic capacity.

Radiobiological parameters: Based on radiobiological models [50], the linear model is often used for in vitro experiments with carbon ion irradiation corresponds to the following formula: *SF* = exp (−α*D*). In preliminary analyses, all data for CIRT at the 3 LET values and X-ray irradiation were fitted using both linear and LQ models with the JMP software (SAS Institute Inc) and the resulting β parameter was close to 0. Moreover, the values of the α parameter were more reproducible using the linear model. Thus, this latter model we used to fit all the survival curves obtained both CIRT and X-ray irradiation. The radiobiological parameters were calculated from the previous equations: SF2 corresponds to the survival fraction at 2 Gy, D37, and D10 are the doses leading to 37% and 10% survival, respectively. To compare the efficacy of X-rays and carbon ions, relative biological effectiveness (RBE) was calculated as the ratio of D37 X-rays/D37 carbon ions. To determine the oxygen effect on cell survival, the oxygen enhancement ratio (OER) was determined as the ratio of D37 in hypoxia/D37 in normoxia. The sensitization enhancement ratio (SER), defined as the ratio of D37 control cells/D37 infected cells, was also quantified to evaluate the impact of the EPOR signaling on the GB cell sensitivity to CIRT.

### 4.5. Cell Cycle Analysis

At 14 and 72 h following cell exposure to carbon ions (4 Gy), the cell cycle was studied by flow cytometry with the Coulter DNA Prep Reagents kit according to the manufacturer’s instructions (Beckman Coulter SAS). Propidium iodide staining was analyzed using a Gallios flow cytometer (Beckman Coulter SAS) with 20,000 events per determination. Analysis and determination of cell distribution in each phase of the cell cycle were performed using Kaluza Analysis software (Beckman Coulter SAS, Villepinte, France).

### 4.6. Western Blot Analysis 

At 24 h after carbon ion irradiation, cells were lysed with RIPA buffer supplemented with 1 µg/mL protease and phosphatase inhibitors. Proteins (50 µg) were separated by SDS-PAGE and transferred to PVDF membranes. The primary antibodies used were P-ERK1/2 (1/200; Santa Cruz, sc-16982-R), ERK (1/200; Santa Cruz, sc-93), and β-actin (1/1000; Cell Signaling, 8H10D10) and they are incubated overnight at 4 °C. Blots were exposed 1 h at room temperature to peroxidase-linked secondary antibodies (anti-rabbit antibody: 1/10,000; Sigma-Aldrich, A0545 and anti-mouse antibody: 1/5000; Sigma-Aldrich, A8924) and the immunoreactive bands were visualized by enhanced chemiluminescence reagents (Thermo Fisher Scientific). For protein quantification, densitometry of the two bands specific of ERK1 and ERK2, either for the phosphorylated or total proteins, were combined. The P-ERK quantification was normalized to the total ERK as is commonly done. The band intensity of the different proteins was determined by ImageJ software [51]. The whole blots showing all bands with all molecular weight markers are presented in Appendix A.

### 4.7. Statistical Analyzes

All data were presented as mean ± SD. The statistical analyses were performed with STATISTICA (TIBCO Software Inc., Palo Alto, CA, USA). The tests used and the number of experiments are detailed in each figure legend.

## 5. Conclusions

In summary, this in vitro study performed on human GB cells demonstrates that high-LET values of carbon ion beams overcome hypoxia-induced radioresistance but also shows for the first time that, depending on the cell type and the activation status of the ERK signaling pathway, the effectiveness of the CIRT can be reduced in hypoxic conditions. In addition, our results underscore the importance of the signaling pathway of EPO, the HIF target gene, in optimizing the response to CIRT.

## Figures and Tables

**Figure 1 cancers-12-02019-f001:**
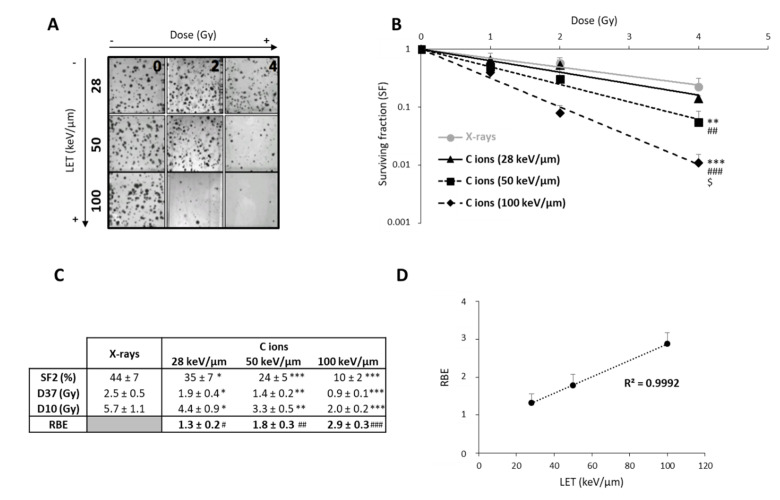
Radiosensitivity of U251 glioblastoma cells as a function of linear energy transfer (LET). (**A**) Representative photographs of U251 colonies obtained 10 days after carbon ion irradiation at 0, 2, and 4 Gy with different LET (28, 50, and 100 keV/µm); (**B**) Survival curves of U251 cells exposed under normoxia (21% O_2_) to X-rays or carbon ions with physical doses ranging from 0 to 4 Gy. Fisher’s LSD post-hoc test after a significant two-way ANOVA (group and dose effects): ** *p* < 0.01, *** *p* < 0.0001 vs. X-rays; ## *p* < 0.01, ### *p* < 0.0001 vs. C ions 28 keV/µm; and $ *p* < 0.0001 vs. C ions 50 keV/µm; (**C**) Comparison of radiological parameters obtained from the fit of survival curves for the different irradiation types. For SF2 (survival fraction at 2 Gy), D37, and D10 (doses leading to 37% and 10% of survival, respectively): * *p* < 0.05, ** *p* < 0.01, *** *p* < 0.0001 vs. X-rays (Fisher’s LSD post-hoc test after a significant one-way ANOVA). For RBE (relative biological effectiveness = ratio of D37 X-rays/D37 carbon ions): # *p* < 0.05, ## *p* < 0.01, ### *p* < 0.0001 vs. theoretical value = 1 (univariate *t*-test); (**D**) Positive correlation between RBE and LET values. All data in this figure represent the mean ± SD of 4 different experiments performed in triplicates (*n* = 3).

**Figure 2 cancers-12-02019-f002:**
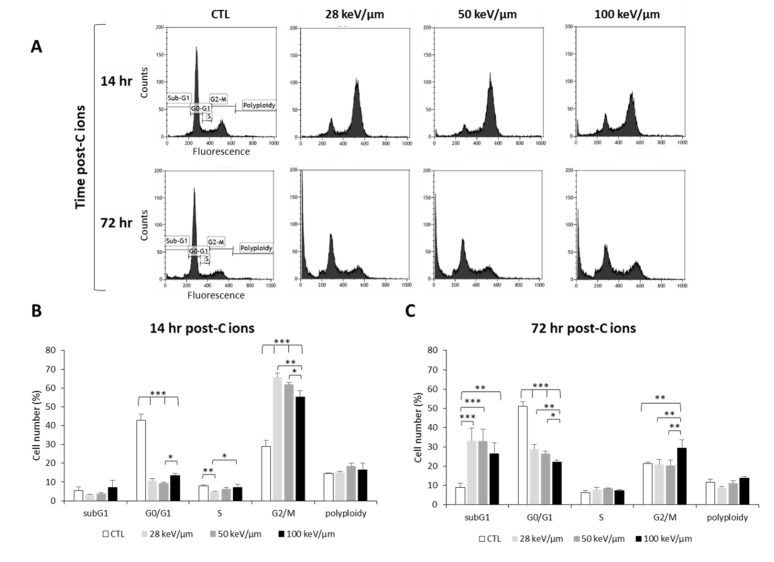
Effect of carbon ion irradiation on the cell cycle of U251 glioblastoma cells. (**A**) Cell cycle profiles of U251 cells exposed under normoxia (21% O_2_) to carbon ions (4 Gy) with various LET (28, 50, and 100 keV/µm) assessed at 14 h and 72 h after irradiation; (**B**) Quantification of the cell distribution in the different phases of the cell cycle at 14 h and (**C**) at 72 h after carbon ion treatment. Mean ± SD, *N* = 3 different experiments for both irradiation conditions. Fisher’s LSD post-hoc test after significant one-way ANOVA; * *p* < 0.05, ** *p* < 0.01, and *** *p* < 0.0001.

**Figure 3 cancers-12-02019-f003:**
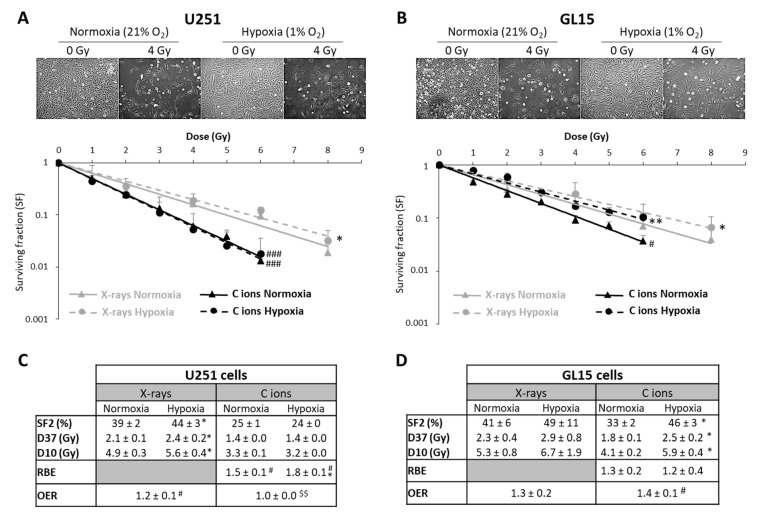
Comparison of survival after exposure of glioblastoma cells to X-ray and carbon ion irradiation in normoxia and hypoxia. (**A**,**B**) Representative photographs of the cell morphology observed 72 h after carbon ion irradiation in normoxia or hypoxia (4 Gy, C ions 28 keV/µm) for U251 cells (A-top) and GL15 cells (B-top). Survival curves from clonogenic assays performed under normoxic (21% O_2_) or hypoxic conditions (1% O_2_) after X-rays or carbon ions (28 keV/µm) for U251 cells (A-bottom) and GL15 cells (B-bottom). Fisher’s LSD post-hoc test after significant two-way ANOVA (group and dose effects): * *p* < 0.05, ** *p* < 0.01 vs. normoxia for X-rays or CIRT; # *p* < 0.05, ### *p* < 0.0001 vs. X-rays in normoxia or hypoxia; (**C**) Quantification of radiobiological parameters obtained after X-rays or CIRT in U251 cells and (**D**) GL15 cells grown under normoxic or hypoxic conditions. SF2, D37, and D10: * *p* < 0.05 vs. normoxia (Fisher’s LSD post-hoc test after a significant one-way ANOVA). RBE: # *p* < 0.05 vs. theoretical value = 1 (univariate *t*-test) and * *p* < 0.05 vs. normoxia (Student’s *t*-test). OER: # *p* < 0.05 vs. theoretical value = 1 (univariate *t*-test) and $$ *p* < 0.01 vs. X-rays (Student’s *t*-test). All data in the figure are presented as mean ± SD of 3 different experiments performed in triplicate (*n* = 3).

**Figure 4 cancers-12-02019-f004:**
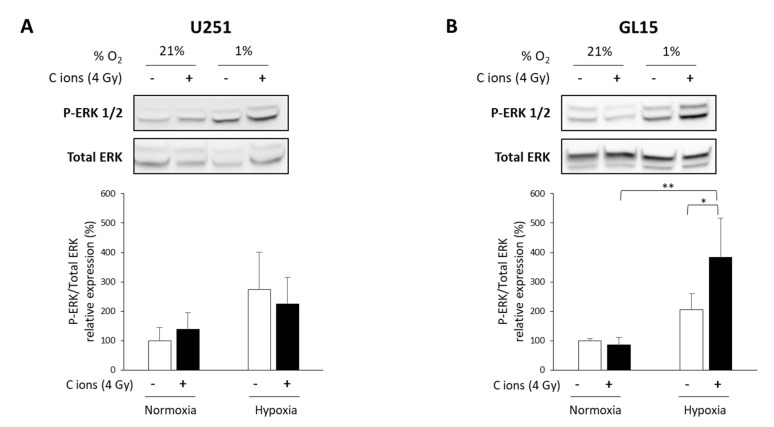
Evaluation of ERK activation after carbon ion irradiation in glioblastoma (GB) cells cultured in normoxic and hypoxic conditions. (**A**,**B**) Representative western blots of phospho-ERK1/2 and total ERK at 24 h after carbon ion irradiation (4 Gy, C ions 28 keV/µm) under normoxic (21% O_2_) or hypoxic conditions (1% O_2_) in U251 cells (A-top) and GL15 cells (B-top). Quantitative analyzes of the ratio P-ERK/Total ERK for U251 cells (A-bottom) and GL15 cells (B-bottom). Mean ± SD, *N* = 3 different samples for each condition. Fisher’s LSD post-hoc test after a significant one-way ANOVA: * *p* < 0.05 and ** *p* < 0.01.

**Figure 5 cancers-12-02019-f005:**
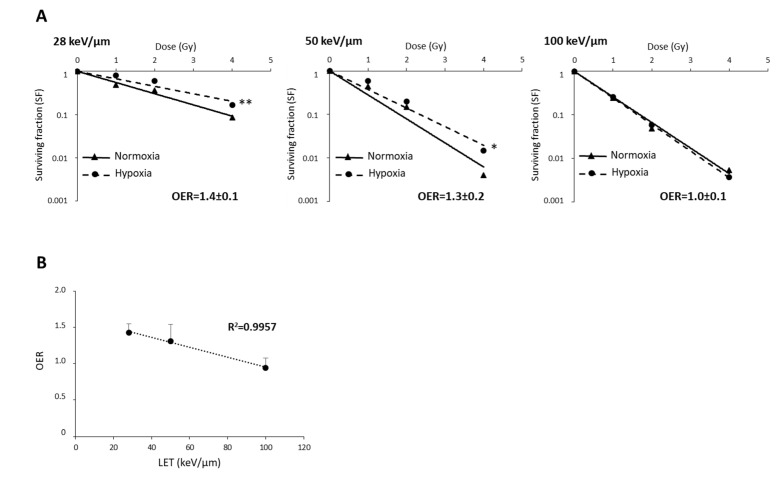
The relationship between oxygen enhancement ratio (OER) and LET of carbon ions in GL15 glioblastoma cells. (**A**) Survival curves of GL15 cells in normoxia (21% O_2_) or hypoxia (1% O_2_) exposed to carbon ion irradiation doses ranging from 0 to 4 Gy at different LET values (28 keV/µm, 50 KeV/µm, and 100 keV/µm). For each LET, OER quantification was performed from D37 determined in normoxia and hypoxia. (**B**) Negative correlation between OER and LET values. Mean ± SD, *N* = 3 different experiments performed in triplicate (*n* = 3) for normoxia and hypoxia conditions. Fisher’s LSD post-hoc test after a significant two-way ANOVA (oxygen and dose effects): * *p* < 0.05, ** *p* < 0.01 vs. normoxia.

**Figure 6 cancers-12-02019-f006:**
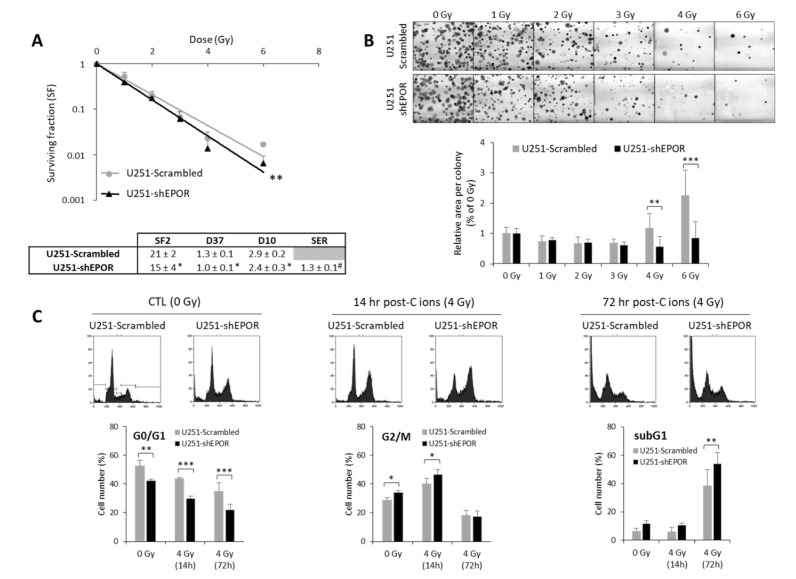
Effect of erythropoietin receptor (EPOR) downregulation on the sensitivity of U251 glioblastoma cells to carbon ion radiotherapy. (**A**) Survival curves of U251-scrambled and U251-shEPOR cells irradiated with carbon ion beams (C ions 50 KeV/µm) under normoxia with doses ranging from 0 to 6 Gy. From clonogenic assays, the radiobiological parameters (SF2, D37, D10) and the sensitization enhancement ratio (D37 scrambled cells/D37 shEPOR cells) were quantified. Mean ± SD, *N* = 4 different experiments performed in triplicate (*n* = 3) for both cell lines. For the survival curves: Fisher’s LSD post-hoc test after a significant two-way ANOVA (group and dose effects) with ** *p* < 0.01 vs. U251-scrambled. For the radiobiological parameters: Student’s *t*-test with * *p* < 0.05 vs. U251-scrambled. For SER (sensitization enhancement ratio = D37 scrambled cells / D37 shEPOR cells): # *p* < 0.05 vs. theoretical value = 1 (univariate *t*-test); (**B**) Representative photographs of U251-scrambled and U251-shEPOR colonies obtained 10 days after carbon ion irradiation at increasing doses (21% O_2_, C ions 50 KeV/µm). Measurement of area per colony for each irradiation dose for U251 cells genetically modified or not. Mean ± SD, *N* = 4 different experiments performed in triplicate (*n* = 3) for both cell lines and irradiation conditions. Fisher’s LSD post-hoc test after a significant one-way ANOVA: ** *p* < 0.01 and *** *p* < 0.0001 vs. U251-scrambled. (**C**) Cell cycle profiles of U251-scrambled and U251-shEPOR cells exposed in normoxia to carbon ions (4 Gy, C ions 50 KeV/µm) obtained at different times after irradiation (0, 14, and 24 h) (C-Top). Quantification of the cell distribution in different phases of cell cycle as a function of time (C-Bottom). Mean ± SD, *N* = 3 different experiments for both cell lines and time after carbon ion irradiation. Fisher’s LSD post-hoc test after a significant one-way ANOVA: * *p* < 0.05, ** *p* < 0.01 and *** *p* < 0.0001.

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
