# Peer review of "Impact of Hypoxia on Carbon Ion Therapy in Glioblastoma Cells: Modulation by LET and Hypoxia-Dependent Genes"

_cancers, 2020, doi:10.3390/cancers12082019_

Round 1
Reviewer 1 Report
The authors have adequately addressed all concerns.
Author Response
We thank the reviewer for agreeing to the changes made to our manuscript.

Reviewer 2 Report
The article by Valable et al. describes the impact of hypoxia on carbon ion therapy for glioma cells. The authors carried out valuable experiments, however, it is difficult to accept the manuscript in the present form. The following issues need to be figured out by the authors before further evaluation:
- As carbon ion beam therapy for malignant glioma is being applied in clinical trials, basic research in this area should be aimed at distinguishing additional mechanisms that can be beneficial for clinical application, and irradiation parameters should comply with clinical application otherwise justified if different. In the introduction, the authors describe carbon ion therapy as high-LET radiotherapy, though in discussion, low-LET (28 keV/µm) and high-LET (100 keV/µm) carbon ion irradiation is compared (lines 270-271). Therefore, it might be confusing for the readers not familiar with CIRT. In the literature, the LET ranging approximately 20–600 keV/μm under aerobic and hypoxic conditions applied to different cell lines has been studied. In this regard, the authors should explain more exactly what LET is considered to be high and what parameters can be used clinically for more effective glioma therapy. Reference: Furusawa Y, Fukutsu K, Aoki M, et al. Inactivation of aerobic and hypoxic cells from three different cell lines by accelerated (3)He-, (12)C- and (20)Ne-ion beams [published correction appears in Radiat Res. 2012 Jan;177(1):129-31]. Radiat Res. 2000;154(5):485-496. https://doi.org/10.1667/0033-7587(2000)154[0485:ioaahc]2.0.co;2
- For X-ray irradiation, the authors used an X-Rad 225 small animal irradiator, that works in keV x-ray range, which is quite different from clinically used MeV-ranged X-ray irradiators, and the effect on biological tissues, including glioma cells, will be different. Despite the number of provided filters for the X-Rad 225 irradiator (with 2mm Al filter as a referenced standard), the authors chose 1mm Cu filter, which cuts most soft X-rays from the provided spectrum, leaving higher energy x-rays that are much less effectively absorbed by tumor cells than, compared to softer X-rays in case of 1mm Al filter, for example. Harder x-rays might be more appropriate for small animals, but when we use cells in such irradiation experiments, the difference in filtration for such a small irradiator might change the effect on cells, and the reason for using a certain filter should be justified. If the authors repeat the irradiation using 1mm Al filter, the cell survival might be different, and the difference between X-rays and the carbon ion beam might be also different simply due to more effective absorption of softer x-rays by tumor cells. The spectrum of x-rays might play a role in the irradiation effect, therefore referencing other experiments regarding glioma cell X-ray irradiation might clarify the issue. The x-ray spectrum calculation has been previously described: Poludniowski G, Landry G, DeBlois F, Evans PM, Verhaegen F. SpekCalc: a program to calculate photon spectra from tungsten anode x-ray tubes. Phys Med Biol. 2009;54(19):N433-N438. doi:10.1088/0031-9155/54/19/N01
- The authors use the word “glioma” in the title and “glioblastoma” further in the text. The authors should unify the terminology, either they reference to malignant gliomas in general, or to specifically one type of gliomas, glioblastoma (grade IV, according to the WHO classification), and such decision should be explained.
- EPO, VEGF, GLUT-1, DNA-PK abbreviations should be explained in full when first mentioned in the text in case some readers are not familiar with them.
- In the introduction (lines 82-83), the authors state that the influence of HIF-1 and/or its dependent genes on tumor sensitivity to CIRT has never been studies, whereas related articles can be found in the internet, making the statement inappropriate. For example: Minami et al. The Effect of Carbon Ion Beam Irradiation for Hypoxia-Mediated Invasion of Glioblastoma, Nano Biomedicine, 2014, Volume 6, Issue 1, Pages 1-11, Released August 24, 2014, Online ISSN 2185-4734, Print ISSN 1883-5198, https://doi.org/10.11344/nano.6.1
- The authors should put the readers’ focus more on new uninvestigated parts of their research. The issue that radiobiological effect of carbon ions on glioblastoma cells is dependent on LET has also been studied and published by other researchers previously, which decreases the novelty of the results of the present study. Reference, for example: Held KD, Kawamura H, Kaminuma T, et al. Effects of Charged Particles on Human Tumor Cells. Front Oncol. 2016;6:23. Published 2016 Feb 12. https://doi.org/10.3389/fonc.2016.00023
- The use of the “basic ballistic” linear quadratic model for current experiments using carbon ion beam might be debatable, as it would be better first show that the “beta” parameter equals zero in their experimental data, and there is no other influence on cell survival rather than carbon ion beam irradiation. The use of the full linear quadratic formula by other authors in related experiments using carbon ion beam irradiation of glioma cells shows exponential SF decrease. The way the authors use the formula should be explained and justified, otherwise the full equation should be applied. Reference, for example: Onishi M, Okonogi N, Oike T, et al. High linear energy transfer carbon-ion irradiation increases the release of the immune mediator high mobility group box 1 from human cancer cells. J Radiat Res. 2018;59(5):541-546. https://doi.org/10.1093/jrr/rry049
- The authors used different LET in different experiments, for example 28 keV/µm in hypoxia experiments and 50 keV/µm in EPO-related experiments. The reason might not be clear for the readers; therefore, this issue should be explained.
- In EPO-related experiments, the authors used U251 cells to evaluate the effect of oxygen, but previously it was shown that hypoxia has no effect on U251 cells that are similarly susceptible to carbon ion irradiation either in normoxia or hypoxia (lines 152-153). The EPO-related study includes only U251 cell line and only 50 keV/µm LET. This is confusing and should be explained.
- The authors should avoid conclusive titles for the figures and the paragraphs. In Figures 3 and 4, for example, the titles tend to summarize the results but actually the conclusion is true only for one cell line, therefore such titles are inappropriate. Figure 3, the hypoxia had influence only to GL-15 cells, in Figure 4, the statement was true for only one GL-15 cell line.
- The article should be proofread to avoid English grammar mistakes (….effects……is dependent…., line 90, etc.) and correct the style when various fonts, font sizes, and reference marks are used, which reminds copying parts of the text from other sources.
- The authors should write the aims of the study more precisely, including the experiments shown in the results (cell cycle, for example), as lack of explanation might be confusing for the readers.
- The conclusion regarding combination of high LET with high dose being irrelevant strategy due to “surprising” results shown in Supplementary Figure S2 and on lines 210-223 is too preliminary and should be avoided, leaving this material to the discussion. And in general, the influence of doses in vitro might not have actual relation to in vivo effects and further clinical application.
- Regarding the colony-forming assay, please use glutaraldehyde for cell fixation, as using ethanol might result in a lot of dust on the surface of the plates, making the images dark and the number of colonies unclear. The presented CF-assay figures are generally dark, and at that size might be difficult to read. CF-assay protocol: Franken, N., Rodermond, H., Stap, J. et al. Clonogenic assay of cells in vitro. Nat Protoc 1, 2315–2319 (2006). https://doi.org/10.1038/nprot.2006.339
- Proper references should be used in the Discussion right after the sentences describing other studies (lines 295-296, 306-308), as the results shown there are unrelated to the current study and therefore add confusion.
- In lines 291-294, the importance of the beta coefficient that represents a relevant parameter of the RBE is described, however, the authors simply withdrew the beta coefficient from the LQ-formula in carbon ion irradiation experiments. Please comment on this.
Carbon ion beam therapy is an alternative modality to treat gliomas, that are characterized by poor prognosis, and to potentially provide patients with additional chances for survival, any study on CIRT and its improvement is of significant value. Taking everything into account, the article should be reevaluated after a major revision.
Author Response
We would like to acknowledge the reviewers for their helpful comments. Please find below a point-by-point response to the reviewers.
Our answers are in blue. Sentences highlighted in grey have been added to the revised manuscript.
Point 1: As carbon ion beam therapy for malignant glioma is being applied in clinical trials, basic research in this area should be aimed at distinguishing additional mechanisms that can be beneficial for clinical application, and irradiation parameters should comply with clinical application otherwise justified if different. In the introduction, the authors describe carbon ion therapy as high-LET radiotherapy, though in discussion, low-LET (28 keV/µm) and high-LET (100 keV/µm) carbon ion irradiation is compared (lines 270-271). Therefore, it might be confusing for the readers not familiar with CIRT. In the literature, the LET ranging approximately 20–600 keV/μm under aerobic and hypoxic conditions applied to different cell lines has been studied. In this regard, the authors should explain more exactly what LET is considered to be high and what parameters can be used clinically for more effective glioma therapy. Reference: Furusawa Y, Fukutsu K, Aoki M, et al. Inactivation of aerobic and hypoxic cells from three different cell lines by accelerated (3)He-, (12)C- and (20)Ne-ion beams [published correction appears in Radiat Res. 2012 Jan;177(1):129-31]. Radiat Res. 2000;154(5):485-496. https://doi.org/10.1667/0033-7587(2000)154[0485:ioaahc]2.0.co;2
We agree with the reviewer’s comment that low-LET and high-LET applied for radiation therapy or CIRT need to be clarified. First, low-LET radiation refers to gamma rays and X-rays with LET values below 5 keV/µm for medical applications whereas high-LET radiation includes alpha particles, protons, neutrons and heavy charged particles, with LET up to 600 keV/µm. This information is now added in the introduction of the revised manuscript (lines 60 to 62).
“Low-LET radiation refers to gamma rays and X-rays, with LET values below 5 keV/µm for medical applications, whereas high-LET radiation includes alpha particles, protons, neutrons and heavy charged particles with LET up to 600 keV/µm.”
In this field of research, the range of LET values for carbon ions is wide with a maximum of 600 keV/µm (Furusawa Y. et al. Radiat. Res. 2000, 154, 485–496). However, using very high-LET values is not suitable as it has been shown that at values over 100 keV/μm, the RBE of carbon ions drops rapidly because the excess dose is not required for cell killing and is transferred to a single cell in the track of a single ion (phenomena called the “overkill effect”) (Jones, B. Front Oncol 2015, 5, 184). Brahme determined that the optimum LET for maximum therapeutic advantage on tumors is between 25 to 75 keV/µm, while minimization of the effects to normal tissues (Brahme A. Int J Radiat Oncol Biol Phys. 2004, 58(2):603-616). Based on these observations, we used in our preclinical study carbon ion LET values consistent with the clinical situation, i.e. LET values ranging between 28 KeV/µm and 100 keV/µm. The carbon ion beam (12C) available to the GANIL facilities has an energy of 95 MeV/u, corresponding to native LET of 28 keV/µm. By using a degrader made PMMA (polymethyl methacrylate) and placed upstream of the culture flasks, we could expose cells to 50 keV/µm (PMMA=13.9 mm) and 100 keV/µm (PMMA=17.9 mm) (see Supplementary Figure S1A). Using these LET values is very interesting to study the biological response of GB cells because the 28 keV/µm LET corresponds to energy deposit upstream of the Bragg peak, 50 keV/µm refers to a deposit energy at the beginning of the Bragg peak and 100 keV/µm is the maximal deposit energy (see Supplementary Figure S1B). To better compare the effect of LET values on the GB response to CIRT, we used the term of “low-LET CIRT” for 28 keV/µm and “high-LET CIRT” for 100 keV/µm. The justification of these 3 LET values used in our study is now detailed at the beginning of the results section (lines 114 to 124).
“Thus, we first investigated the biological response of GB cells to CIRT at LET values ranging from 28 keV/µm to 100 keV/µm. These LET values were chosen to approximate those used in the clinics. Indeed, it has been shown that the optimum LET to maximize the therapeutic advantage on tumors is between 25 to 75 keV/µm, while minimization of the effects to normal tissues [32]. The maximal LET must not exceed 100 keV/µm because over this value, the RBE of carbon ions drops rapidly as the dose in excess not necessary for cell killing is transferred to a cell in the track of a single ion (phenomena called the “overkill effect”) [33]. In this study, 28 keV/µm (defined as low-LET CIRT) corresponds to the energy deposit upstream of the Bragg peak, 50 keV/µm refers to a deposit energy at the beginning of the Bragg peak and 100 keV/µm (corresponds to high-LET CIRT) is the deposit energy maximal (without induce overkill effect).”
Point 2: For X-ray irradiation, the authors used an X-Rad 225 small animal irradiator, that works in keV x-ray range, which is quite different from clinically used MeV-ranged X-ray irradiators, and the effect on biological tissues, including glioma cells, will be different. Despite the number of provided filters for the X-Rad 225 irradiator (with 2mm Al filter as a referenced standard), the authors chose 1mm Cu filter, which cuts most soft X-rays from the provided spectrum, leaving higher energy x-rays that are much less effectively absorbed by tumor cells than, compared to softer X-rays in case of 1mm Al filter, for example. Harder x-rays might be more appropriate for small animals, but when we use cells in such irradiation experiments, the difference in filtration for such a small irradiator might change the effect on cells, and the reason for using a certain filter should be justified. If the authors repeat the irradiation using 1mm Al filter, the cell survival might be different, and the difference between X-rays and the carbon ion beam might be also different simply due to more effective absorption of softer x-rays by tumor cells. The spectrum of x-rays might play a role in the irradiation effect, therefore referencing other experiments regarding glioma cell X-ray irradiation might clarify the issue. The x-ray spectrum calculation has been previously described: Poludniowski G, Landry G, DeBlois F, Evans PM, Verhaegen F. SpekCalc: a program to calculate photon spectra from tungsten anode x-ray tubes. Phys Med Biol. 2009;54(19):N433-N438. doi:10.1088/0031-9155/54/19/N01
We fully agree with the reviewer’s comment. We are aware of this limitation and have previously evaluated by clonogenic assays the difference in GB cell survival (U251 cells) between the preclinical irradiator (XRAD 225 Cx with 1mm Cu filter; energy 80 keV) and a clinical irradiator (linear accelerator Therac-15 Saturne; energy 15 MeV). As presented on the Figure A (see PDF attached), the profiles of survival curves for U251 cells are similar, independently of the irradiator type.
In our laboratory, we have also conducted preclinical studies on glioblastoma models (such as U251 models inoculated into nude rats and mice) (Corroyer-Dulmont A et al., Eur J Nucl Med Mol Imaging. 2016;43(4):682-694; Pérès EA et al., Oncotarget. 2015;6(4):2101-2119). For animal models, it is important to use 1mm Cu filter that only let through hard X-rays to ensure optimal absorption of X-rays in depth. Thus, it was necessary for us to keep 1mm Cu filter to elucidate in vitro the cellular mechanisms at play in the tumor response to irradiation observed in in vivo models. Moreover, it seems to be more appropriate to use a Cu filter with an orthovoltage irradiator to better match clinical conditions: selecting only hard X-rays with the Cu filter (1mm) allows having more penetrating radiation, although the energy of the radiation beam is low.
Point 3: The authors use the word “glioma” in the title and “glioblastoma” further in the text. The authors should unify the terminology, either they reference to malignant gliomas in general, or to specifically one type of gliomas, glioblastoma (grade IV, according to the WHO classification), and such decision should be explained.
As requested by the reviewer, we have replaced the term “glioma” by “glioblastoma” in the title as well as in the text.
Point 4: EPO, VEGF, GLUT-1, DNA-PK abbreviations should be explained in full when first mentioned in the text in case some readers are not familiar with them.
The abbreviations for these molecules have now been incorporated in the revised version of the manuscript (lines 96-97).
“as EPO (erythropoietin) [24], VEGF (vascular endothelial growth factor) [25], GLUT-1 (glucose transporter 1) [26] or DNA-PK (DNA-dependent protein kinase) [27].”
Point 5: In the introduction (lines 82-83), the authors state that the influence of HIF-1 and/or its dependent genes on tumor sensitivity to CIRT has never been studies, whereas related articles can be found in the internet, making the statement inappropriate. For example: Minami et al. The Effect of Carbon Ion Beam Irradiation for Hypoxia-Mediated Invasion of Glioblastoma, Nano Biomedicine, 2014, Volume 6, Issue 1, Pages 1-11, Released August 24, 2014, Online ISSN 2185-4734, Print ISSN 1883-5198, https://doi.org/10.11344/nano.6.1
We agree with the reviewer that there are a few studies investigating sensitivity to CIRT that have been published (Minami K et al., Nano Biomedicine 6(1),1-11; Chiblak S et al., JCI Insight. 2019;4(2):e123837). We apologize for the oversight and are now citing these studies in the revised version of the manuscript (line 100).
“Although the modulation of HIF-1 expression after exposure to carbon ions has been evaluated [28,29], the influence of HIF-1 and/or HIF-dependent genes on the intrinsic sensitivity of tumor cells to CIRT is poorly studied [30,31].”
Point 6: The authors should put the readers’ focus more on new uninvestigated parts of their research. The issue that radiobiological effect of carbon ions on glioblastoma cells is dependent on LET has also been studied and published by other researchers previously, which decreases the novelty of the results of the present study. Reference, for example: Held KD, Kawamura H, Kaminuma T, et al. Effects of Charged Particles on Human Tumor Cells. Front Oncol. 2016;6:23. Published 2016 Feb 12. https://doi.org/10.3389/fonc.2016.00023
As described at the end of the introduction section focus (lines 106-109), the main objective of the study is to increase current knowledge about the impact of hypoxia on GB sensitivity to CIRT. Although numerous radiobiology studies have shown that RBE is correlated with LET and OER is reduced by increasing the LET values ​​of carbon ions, few studies have focused on glioblastoma where tumors are very resistant to low-LET radiation, in particular because of their hypoxic environment. Hence, in our study, we were interested to study GB cell response to CIRT in presence or not of hypoxia. We investigated whether CIRT efficacy might be dependent on the GB cell line studied, or associated with the ERK pathway and whether a HIF-dependent genes (such as EPOR signaling) might interfere in GB sensitivity to CIRT. We now re-write the end of the introduction to clarify our aims (lines 106-109).
“The aim of this in vitro study was to determine the impact of oxygen levels on the sensitivity of GB cells to CIRT. We used human GB cells lines to determine: i) whether carbon ion irradiation can overcome hypoxia-induced radioresistance; and ii) whether varying LET values or downregulating the EPO receptor (EPOR) can modulate GB cell sensitivity to carbon ions under hypoxic conditions.”
Point 7: The use of the “basic ballistic” linear quadratic model for current experiments using carbon ion beam might be debatable, as it would be better first show that the “beta” parameter equals zero in their experimental data, and there is no other influence on cell survival rather than carbon ion beam irradiation. The use of the full linear quadratic formula by other authors in related experiments using carbon ion beam irradiation of glioma cells shows exponential SF decrease. The way the authors use the formula should be explained and justified, otherwise the full equation should be applied. Reference, for example: Onishi M, Okonogi N, Oike T, et al. High linear energy transfer carbon-ion irradiation increases the release of the immune mediator high mobility group box 1 from human cancer cells. J Radiat Res. 2018;59(5):541-546. https://doi.org/10.1093/jrr/rry049
We acknowledge the reviewer’s pertinent comment, which led us to realize that we introduced a mistake when writing the paragraph on "Radiobiological parameters" in the Material and Methods section (lines 480 to 486).
To properly determine the radiobiological parameters, we have to check beforehand which radiobiological model is the most suitable for better fitting the survival fraction data as a function of the irradiation dose. While there are many mathematical modelling the action of radiation on living cells (Bodgi L et al., Theor Biol. 2016;394:93-101), the most often used are a linear model, referring to “the single-target single-hit theory” (SF=exp (−αD− βD2)) and a linear quadratic model (LQ model) corresponding to “double hit single target theory” (SF=exp (−αD)). As presented in Figure B-item A (see PDF attached), the fitting of the survival curves with linear or LQ models are similar for CIRT at all LET values studied and with X-ray irradiation. From the data presented in Figure 1 of the manuscript, the quantification of the α and β parameters with the LQ model shows β values very close to 0 and a large variability of α values (Figure B-item B) (see PDF attached). The linear model gives more reproducible α values between the different experiments (N = 4 for each irradiation type) (Figure B-item B) (see PDF attached). Based on these results, we retained the linear model to fit all the survival curves obtained with both CIRT and X-ray irradiation and to extract the radiobiological parameters (SF2, D37, D10, RBE and OER). The explanation on the choice of the model used has now been added in the Material and Methods section (lines 480 to 486).
“Based on radiobiological models [50], the linear model is often used for in vitro experiments with carbon ion irradiation corresponds to the following formula: SF=exp (−αD). In preliminary analyses, all data for CIRT at the 3 LET values and X-ray irradiation were fitted using both linear and LQ models with the JMP software (SAS Institute Inc) and the resulting β parameter was close to 0. Moreover values the α parameter were more reproducible using the linear model. Thus, this latter model we used to fit all the survival curves obtained both CIRT and X-ray irradiation.”
Point 8: The authors used different LET in different experiments, for example 28 keV/µm in hypoxia experiments and 50 keV/µm in EPO-related experiments. The reason might not be clear for the readers; therefore, this issue should be explained.
For the experiments comparing normoxic and hypoxic conditions, we first presented the results obtained with LET value of 28 keV/µm because the OER difference between U251 and GL15 cells was more pronounced (Figure 3) but similar data are presented for other LET values in Figure 5 (for GL15 cells) and Supplementary Figure S2 (for U251 cells). In the regard to the experiments combining EPOR inhibition and CIRT (Figure 6), we used a LET value of 50 keV/µm in order to match the optimum LET range used in clinical practice (25–75 keV/µm) and to avoid the overkill effect usually described over 100 keV/µm for carbon ions. The choice of LET value (50 keV/µm) for experiments evaluating the EPOR knockdown on the CIRT efficacy is now justified in the revised manuscript (lines 288 to 291).
“In these experiments, we chose to use an LET value of 50 keV/µm in order to match the optimum LET range used in clinical practice (25–75 keV/µm) and to avoid the overkill effect usually associated with LET values > 100 keV/µm.”
Point 9: In EPO-related experiments, the authors used U251 cells to evaluate the effect of oxygen, but previously it was shown that hypoxia has no effect on U251 cells that are similarly susceptible to carbon ion irradiation either in normoxia or hypoxia (lines 152-153). The EPO-related study includes only U251 cell line and only 50 keV/µm LET. This is confusing and should be explained.
We totally agree with the reviewer's comment and it is a limitation of our study. When we initially performed these experiments, the stable EPOR knockdown GL15 cell line was not available yet. We had therefore planned to perform new experiments at the end of March 2020 to determine the effects of EPOR knockdown in GL15 cells on carbon ion efficacy in normoxia and hypoxia (1% O2). Unfortunately, due to the COVID-19 pandemic, access to the GANIL facilities was postponed until late 2021. Thus, we cannot provide additional experimental evidence to demonstrate the implication of EPO/EPOR signaling in hypoxia-induced radioresistance with carbon irradiation. We are aware that it represents a limitation of our study, which is now pointed out in the discussion section (lines 427 to 430).
“These results were obtained under normoxia, and it would be interesting in future experiments to investigate the involvement of EPO/EPOR signaling in hypoxia-induced resistance to carbon ion irradiation. It would also be suitable to evaluate the effects of EPOR inhibition in GL15 cells, a cell line that exhibits hypoxia-dependent radioresistance to CIRT.”
Point 10: The authors should avoid conclusive titles for the figures and is them the paragraphs. In Figures 3 and 4, for example, the titles tend to summarize the results but actually the conclusion is true only for one cell line, therefore such titles are inappropriate. Figure 3, the hypoxia had influence only to GL-15 cells, in Figure 4, the statement was true for only one GL-15 cell line.
In the results section, titles for all paragraphs are now modified at lines 112, 181 and 283. As recommended by the reviewer, we have made changes to the titles of Figure 1 (line 139), Figure 2 (line 174), Figure 3 (line 210), Figure 4 (line 241) and Figure 6 (line 310).
Point 11: The article should be proofread to avoid English grammar mistakes (….effects……is dependent…., line 90, etc.) and correct the style when various fonts, font sizes, and reference marks are used, which reminds copying parts of the text from other sources.
The revised manuscript has been proofread by a professional Editor. We have also edited the manuscript to enhance clarity.
Point 12: The authors should write the aims of the study more precisely, including the experiments shown in the results (cell cycle, for example), as lack of explanation might be confusing for the readers.
As we mentioned in the response to comment 5 and in order to be more focused, we rewrote a paragraph at the end of the introduction (lines 106 to 109) and at the end of the conclusion (lines 528 to 532).
- In the Introduction section (lines 106 to 109):
“The aim of this in vitro study was to determine the impact of oxygen levels on the sensitivity of GB cells to CIRT. We used human GB cells lines to determine: i) whether carbon ion irradiation can overcome hypoxia-induced radioresistance; and ii) whether varying LET values or downregulating the EPO receptor (EPOR) can modulate GB cell sensitivity to carbon ions under hypoxic conditions.”
- In the Conclusion section (lines 528 to 532):
“In summary, this in vitro study performed on human GB cells demonstrates that high-LET values ​​of carbon ion beams overcome hypoxia-induced radioresistance but also shows for the first time that, depending on the cell type and the activation status of the ERK signaling pathway, the effectiveness of the CIRT can be reduced in hypoxic conditions. In addition, our results underscore the importance of the signaling pathway of EPO, the HIF target gene, in optimizing the response to CIRT.”
Point 13: The conclusion regarding combination of high LET with high dose being irrelevant strategy due to “surprising” results shown in Supplementary Figure S2 and on lines 210-223 is too preliminary and should be avoided, leaving this material to the discussion. And in general, the influence of doses in vitro might not have actual relation to in vivo effects and further clinical application.
In agreement with the reviewer's comment, we removed this result (Supplementary Figure S2 in the initial manuscript) and its comment (lines 266 to 282 in results section and lines 355 to 373 in discussion section of the initial manuscript), these results being indeed a little too preliminary.
Point 14: Regarding the colony-forming assay, please use glutaraldehyde for cell fixation, as using ethanol might result in a lot of dust on the surface of the plates, making the images dark and the number of colonies unclear. The presented CF-assay figures are generally dark, and at that size might be difficult to read. CF-assay protocol: Franken, N., Rodermond, H., Stap, J. et al. Clonogenic assay of cells in vitro. Nat Protoc 1, 2315–2319 (2006). https://doi.org/10.1038/nprot.2006.339
We would like to thank the reviewer for this insightful comment because we were not aware of this limitation regarding cell fixation for clonogenic assays. We will definitely switch to glutaraldehyde for the next experiments. We never had problems to identify the colonies when fixed with 1% crystal violet diluted in 20% ethanol. The contrast between the colonies and the plastic is adequate (see Figure C), probably due to repeated washing with water after colony fixation. The images of colonies presented in the manuscript seem to be dark, but this effect is due to the digitization of the culture flasks to clearly distinguish the colonies using the macro “cell counter” available in the ImageJ software. As presented in Figure C (see PDF attached), when we adapted the image contrast, the image quality was better. Therefore, the image quality is improved in the Figure 6B in the revised manuscript.
Point 15: Proper references should be used in the Discussion right after the sentences describing other studies (lines 295-296, 306-308), as the results shown there are unrelated to the current study and therefore add confusion.
To avoid confusion between our results and data from literature (lines 381-384), we inserted the reference 43 at the end of the sentence. For the study cited in the lines 372-373 (reference 42), we deleted this paragraph from the discussion (see the response to the item 13).
Point 16: In lines 291-294, the importance of the beta coefficient that represents a relevant parameter of the RBE is described, however, the authors simply withdrew the beta coefficient from the LQ-formula in carbon ion irradiation experiments. Please comment on this.
We totally agree with the reviewer's comment. To avoid confusion, the paragraph that included these sentences has been deleted (see the response to the items 7 and 13).

Round 2
Reviewer 2 Report
The authors did a good job and answered all questions and responded to comments. The article describes an interesting basic study which deserves publishing.
Some minor comments:
Not really critical, but: Line 13: "Glioblastoma (GB) are resistant to low-LET radiation (X-rays), due in part to the hypoxic environment in these brain tumors." might be "Glioblastomas are"(and no further change needed), or "Glioblastoma is" (and the latter part should be also changed), as the authors use "Glioblastoma .... is" further in the text (Line 28).
Critical: Line 28: "Glioblastoma (GB), also defined a grade IV glioma based on the WHO classification,..." should be "defined as".
Line 91: "We used human GB cells lines to determine... ", where "human GB cell lines" might sound better.
Line 383: "Moreover values the α parameter were more reproducible using the linear model." This sentence should be corrected by proper punctuation and, probably, a preposition."
I have no further comments and suggest article acceptance as the authors wrote an impressive reply.
Sincerely,
Author Response
We would like to acknowledge the reviewer for their helpful comments. Please find below a point-by-point response to the reviewers.
Our answers are in blue. The corrections made in the revised manuscript are highlighted in yellow.
Point 1: Not really critical, but: Line 13: "Glioblastoma (GB) are resistant to low-LET radiation (X-rays), due in part to the hypoxic environment in these brain tumors." might be "Glioblastomas are"(and no further change needed), or "Glioblastoma is" (and the latter part should be also changed), as the authors use "Glioblastoma .... is" further in the text (Line 28).
As recommended by the reviewer, we made the changes in the line 13:
“Glioblastoma (GB) is resistant to low-LET radiation (X-rays), due in part to the hypoxic environment in this brain tumor.”
Point 2: Critical: Line 28: "Glioblastoma (GB), also defined a grade IV glioma based on the WHO classification,..." should be "defined as".
We corrected the sentence at line 28 as requested by the reviewer:
“Glioblastoma (GB), also defined as a grade IV glioma based on the WHO classification, is the most common malignant primary brain tumor in adults…”
Point 3: Line 91: "We used human GB cells lines to determine... ", where "human GB cell lines" might sound better.
The error at line 91 has been corrected:
“We used human GB cell lines to determine…”
Point 4: Line 383: "Moreover values the α parameter were more reproducible using the linear model." This sentence should be corrected by proper punctuation and, probably, a preposition."
We done the modification at the line 383:
“Moreover, the values of the α parameter were more reproducible using the linear model.”

This manuscript is a resubmission of an earlier submission. The following is a list of the peer review reports and author responses from that submission.
Round 1
Reviewer 1 Report
This manuscript entitled impact of hypoxia on carbon ion therapy in glioma cells: modulation by LET and hypoxia dependent gens is focusing on the impact of carbon ion therapy in GBM cells. The paper is focusing on cells which are bearing p53 mutations and a EGFR modification. First of all, most of the first and end part of the manuscript are only done in normoxic conditions. It should be completed for the same manipulations in hypoxic conditions too. Furthermore, they should explain in all manipulations why they are focusiong on this subtype of GBM cells and why they are culturing them with a high level of serum which is known to differentiate the cells. What is the status of PTEN in those cells ?
In the first part, the authors are going rapidly in the text to say finally that carbon ion is better than Xray irradiation as the differences especially in the RBE are only observed with a very high LET value. They should maybe focused only on this very high dose and do systematically the same experiments in hypoxic conditions. The same remarks can be applied for the results showed in fugure 2.
For the results focusing on hypoxia showed in the figure 3, the differences are as in previous experiments similar in U251 cells and a slight difference is showed in GL15. They seem to be no oxygen effect in both conditions. They should go further to understand why. Did they do any gene expression analyses ?
For the figure 4, we can not understand why suddenly they are focusing only on ERK as hypoxia is usually explored by the study of HIF1, ERK, pERK, AKT, pAKT, mTorC1 and mTorC2 and further down the downstream signaling proteins. Especially, we can not understand why thereafter they are using EPO as we have no information on EPO itself and HIF1 protein expression. On the western blot showed in the figure 4, we can see 2 bands in the phospho-ERK1/2. The final calculation was then done on which one ? The evaluation od the protein expression was performed after 24h, but they studied the effect of X-ray and carbon ion usually at 14 and 72h. Why did they do their ERK expression at 24h ? is this a specific time point ? I did not see any control protein in the western blot analyses.
Finally, for the EPO manipulation, they did it in a probable normoxic condition, it is not clearly detailed in the manuscript. Why did they only the experiment in one condition and not in both normoxic and hypoxic conditions ?
Reviewer 2 Report
Summary:
This study investigated the effect of LET and hypoxia on treatment with carbon-ion therapy in two glioblastoma cell-lines. They found this to be the case in a cell-line dependent manner related to levels of ERK phosphorylation and hypoxia related genes such as EPO.
Major concerns:
1) Increase in Sub-G0/G1 fraction is indicative of cell death and not necessarily apoptosis. Please use cleaved-CASP3 or cleaved-PARP FACS/western blot or protease activity assay (i.e. Caspase-Glo) to show that the mode of death is indeed apoptosis.
2) The association between radiation resistance and phospho-ERK is intriguing. However, causation is currently speculative. Please perform ERK or MEK inhibition experiments or shRNA knockdown to show decrease in resistance with ERK inhibition.